# Underestimated Aspect of Mucopolysaccharidosis Pathogenesis: Global Changes in Cellular Processes Revealed by Transcriptomic Studies

**DOI:** 10.3390/ijms21041204

**Published:** 2020-02-11

**Authors:** Lidia Gaffke, Karolina Pierzynowska, Magdalena Podlacha, Dżesika Hoinkis, Estera Rintz, Joanna Brokowska, Zuzanna Cyske, Grzegorz Wegrzyn

**Affiliations:** 1Department of Molecular Biology, University of Gdansk, Wita Stwosza 59, 80-308 Gdansk, Poland; lidia.gaffke@phdstud.ug.edu.pl (L.G.); karolina.pierzynowska@biol.ug.edu.pl (K.P.); magdalena.podlacha@biol.ug.edu.pl (M.P.); estera.rintz@gmail.com (E.R.); joanna.brokowska@phdstud.ug.edu.pl (J.B.); zuzia.cyske@op.pl (Z.C.); 2Intelliseq Ltd., Chabrowa 12/3, Stanisława Konarskiego 42/13, 30-046 Cracow, Poland; dzesika.hoinkis@intelliseq.pl

**Keywords:** mucopolysaccharidoses, transcriptomic analyses, cellular processes

## Abstract

Mucopolysaccharidoses (MPS), a group of inherited metabolic disorders caused by deficiency in enzymes involved in degradation of glycosaminoglycans (GAGs), are examples (and models) of monogenic diseases. Accumulation of undegraded GAGs in lysosomes was supposed to be the major cause of MPS symptoms; however, their complexity and variability between particular types of the disease can be hardly explained by such a simple storage mechanism. Here we show that transcriptomic (RNA-seq) analysis of the material derived from fibroblasts of patients suffering from all types and subtypes of MPS, supported by RT-qPCR results, revealed surprisingly large changes in expression of genes involved in various cellular processes, indicating complex mechanisms of MPS. Although each MPS type and subtype was characterized by specific changes in gene expression profile, there were genes with significantly changed expression relative to wild-type cells that could be classified as common for various MPS types, suggesting similar disturbances in cellular processes. Therefore, both common features of all MPS types, and differences between them, might be potentially explained on the basis of changes in certain cellular processes arising from disturbed regulations of genes’ expression. These results may shed a new light on the mechanisms of genetic diseases, indicating how a single mutation can result in complex pathomechanism, due to perturbations in the network of cellular reactions. Moreover, they should be considered in studies on development of novel therapies, suggesting also why currently available treatment methods fail to correct all/most symptoms of MPS. We propose a hypothesis that disturbances in some cellular processes cannot be corrected by simple reduction of GAG levels; thus, combined therapies are necessary which may require improvement of these processes.

## 1. Introduction

Although each of about 10,000 monogenic diseases, described to date [1], is caused by mutations in a single gene, symptoms of the vast majority of them are complex. Therefore, although the primary defect of most of these disorders can be identified, precise molecular mechanisms that lead to specific symptoms remain significantly less recognized. Recent studies, in which almost 600,000 genomes of healthy persons were analyzed, led to the discovery of 13 cases of adults with no symptoms in which mutations of confirmed pathological effects occurred and that should result in manifestation of severe diseases in childhood [2]. While these results demonstrate a possibility of compensation of effects of particular mutations, giving hope for developing novel therapies, they also indicate how far we are from understanding molecular processes leading to the development of particular genetic diseases.

Lysosomal storage diseases (LSD) are often considered model genetic disorders due to the level of understanding of their molecular mechanisms and the stage of development of therapies [3]. Mucopolysaccharidoses (MPS) is a group of intensively investigated LSD caused by deficiency of lysosomal enzymes involved in degradation of glycosaminoglycans (GAG) [4]. Although there are various therapeutic options for MPS, including enzyme replacement therapy, substrate reduction therapy, and hematopoietic stem cell transplantation, none of them provide a way to correct all, or even most, symptoms in humans [5,6].

Impaired hydrolysis of GAG, caused by mutations in genes’ coding for specific lysosomal enzymes, leads to continuous accumulation and storage of these compounds in patients’ cells, causing damage of the affected tissues, including the heart, respiratory system, bones, joints, and central nervous system (CNS). MPS are usually fatal diseases, with an average expected life span of one or two decades, though patients with milder forms can survive into adulthood [4]. On the basis of the nature of lacking an enzyme, or having a defective enzyme, and the kind(s) of stored GAG(s), 11 types and subtypes of MPS are distinguished (Table 1).

A few decades ago, lysosomal GAG storage was considered not only the primary, but also the only, cause of MPS [7]. Although subsequent observations and experimental studies clearly indicated that secondary and tertiary biochemical and cellular changes can significantly contribute to MPS pathology (for summary and comprehensive discussions, see [4]), it was considered a surprise that enzyme replacement therapy, which consists of intravenous administration of recombinant human enzyme that is lacking in the patient’s cells, can only partially correct disease symptoms, despite lowering GAG amounts to the normal level. In fact, even knowing about inefficient crossing of the blood–brain barrier by the enzyme and remembering that the bones are poorly irrigated organs, one should expect higher efficacy of this therapy [8]. Only recent reports have indicated that, in MPS cells, there are significant changes in crucial cellular processes. However, there is a relatively low number of such reports, and they concern specific cases or single types/subtypes of MPS. On the other hand, on the basis of the published results, one can conclude that MPS consists of a much broader spectrum of cellular defects than it was supposed previously. Specifically, proteomic and transcriptomic studies on the material isolated from brains of MPS I and MPS VII animal models indicated that expression of genes coding for proteins involved in formation of cytoskeleton and in vacuolar transport is changed relative to wild-type animals [9,10]. Another report indicated that the cell cycle can be impaired in MPS cells, particularly in MPS II [11]. In the presence of storage material in lysosomes, these organelles cannot function properly; however, studies on specific changes of lysosomal functions were performed only in a few MPS types, as summarized previously [12]. Recent studies indicated that, in the brain tissue withdrawn from MPS IIIC mouse, there are dysfunctions of mitochondrial enzymes [13]. The disturbance of apoptosis and autophagy processes has been recognized as one of the crucial defects leading to cellular dysfunctions in many diseases, including LSD [12]; however, studies on apoptosis and autophagy are infrequent in the case of MPS, and concerned to date only a few MPS types [14,15,16].

In light of the above facts, considering unknown detailed mechanisms leading to specific MPS symptoms, as well as significant differences between particular MPS types, which are unlikely to occur due to simple storage of various GAGs in lysosomes, we aimed to perform complex studies on changes in cellular processes occurring in cells of all types of MPS. We have performed a complex transcriptomic study, and in fact, this is the first report describing simultaneous transcriptomic analyses of all MPS types and subtypes. Our results indicated multiple and significant changes in expression of many genes which have products involved in various cellular processes, shedding a new light on molecular and cellular mechanisms of MPS.

## 2. Results

In this study we analyzed cells (fibroblasts) of all known MPS types (types I, II, IIIA, IIIB, IIIC, IIID, IVA, IVB, VI, VII, and IX) (Table 2) and wild-type counterparts (the human dermal fibroblast, HDFa, and cell line). They were cultured in vitro, and total mRNA was isolated and subjected to transcriptomic analysis, employing RNA sequencing (RNA-seq), as described in Materials and Methods (Section 4). Four biological repeats were performed for each MPS type and the HDFa cell line (by investigating four independent cell cultures, each derived from a different cell passage). Number of reads in each experiment was between 40,457,828 and 61,670,273 (Appendix A), indicating appropriate quality of the biological material and quality of sequencing.

We found that a number of transcripts with significantly changed levels (false discovery rate (FDR) < 0.1; *p* < 0.1) in MPS cells vs. HDFa control varied in different MPS types, and was between 289 (in MPS VI) and 893 (in MPS IVB) (Figure 1). Both up-regulated and down-regulated genes could be found in each MPS type. These results indicate significant changes in expression of several hundred genes in each MPS type, suggesting a reason for global changes in cellular physiology.

In the next step, we asked what cellular processes can be potentially changed due to variation in levels of particular transcripts in different MPS types. Therefore, we have grouped the changed transcripts by ascribing them to genes coding for products involved in particular cellular processes (classified according to QuickGO database). For this analysis, all direct children terms of GO:0009987 (cellular process) were chosen. A direct child term is a more specific GO term that is immediately preceded by the given GO term in the directed acyclic graph (DAG) defined by the Gene Ontology Consortium. The analysis was performed by keeping two values of the FDR, namely FDR < 0.1 (Figure 2, Appendix A) and FDR < 10^−6^ (Appendix A). Among all considered processes, some revealed a particularly high number of changed transcripts. They include the following processes: cell cycle (GO:0007049), cell communication (GO:0007154), signal transduction (GO:0007165), positive and negative regulation of cellular processes (GO:0048522; GO:0048523), cellular development (GO:0048869), cell division (GO:0051301), cellular homeostasis (GO:0019725), actin filament-based processes (GO:0030029), processes utilizing autophagic mechanism (GO:0061919), cell growth (GO:0016049), cell death (GO:0008219), protein folding (GO:0006457), vesicle targeting (GO:0006903), microtubule-based processes (GO:0007017), and many others (all GO terms are provided in Appendix A). Importantly, differences between MPS types were also evident, suggesting that particular processes may be disturbed to various extents in different disease types and subtypes, and this inference is supported by different patterns of heatmaps and numbers of significantly changed transcripts (Appendix A). These results indicate that, apart from a few processes which were reported to be disturbed in some MPS types previously [9,10,11,12,13,14], there are severe cellular changes in a large number of processes which are significantly more pronounced and expanded than expected. In addition, differences between various MPS types appear considerable, pointing to molecular mechanisms of development of different symptoms in the individual diseases from this group that cannot be explained solely on the basis of lysosomal storage of GAG. One can easily assume that specific symptoms characteristic for MPS, like short stature, as well as functional impairment of various cells, tissues, and organs (see [4] for details) can be explained by perturbations in specific cellular processes, requiring products of genes that are unproperly expressed in MPS cells (compare Figure 2, Appendix A; see Discussion for detailed analysis).

To find processes potentially changed to the highest extent, we have analyzed those in which at least 30 kinds of transcripts were particularly strongly changed (FDR < 10^−6^, *p* < 0.1) in different MPS types. The results presented in Figure 3 indicate that there are several processes which may be significantly affected in many, if not all, MPS types, showing plausible common mechanisms operating in MPS as a coherent group. Such great changes in transcripts of genes involved in cellular metabolic processes were detected in all MPS types; those involved in cell communication, cellular component organization, and cellular response to stimulus occurred in all types but MPS IVA; those involved in signal transduction were present in all types but MPS IVA and MPS VI; and those involved in positive and negative regulations were evident in all types but MPS II, MPS IVA, and MPS VI. Drastic changes in expression of genes involved in these processes may explain, at least partially, the reason of severity of all MPS types as disturbances in such basic regulatory pathways should result in severe cell dysfunctions and, thus, abnormalities in tissues and organs, causing severe symptoms in patients.

To compare the number of genes with changed expression in particular MPS types, we have constructed Chord graphs for the processes described above. As indicated in Figure 4, groups of genes with changed expression which are common between each two MPS types could be found, showing significant similarity among MPS types and possible general trends in cellular changes. However, there are some particularly high similarities in individual processes between individual MPS types.

In the GO-termed cellular metabolic processes, the highest number of common genes in which expression was changed occurred between MPS IX and MPS IIIC, MPS IVB and MPS IIIA, MPS IVB and MPS IIIC, and MPS IX and MPS IVB; and, to some extent, MPS VII and MPS IIIA, MPS VII and MPS IIIB, MPS VII and MPS IIIC, MPS IX and MPS IIIA, MPS IX and MPS I, MPS IX and MPS IVB, and MPS IIIA and MPS IIIB.

In cell communication, the highest similarities were between MPS IIIA and MPS IVB, MPS IIIB and MPS IVB, MPS IIIC and MPS IVB, MPS IX and MPS IIIA, MPS IX and MPS IIIB, MPS IX and MPS IIIC, MPS IX and MPS IVB, and MPS IIIA and MPS IIIB.

In cellular component organization, the highest similarities were between the following: MPS IIIA and MPS IVB; MPS IIIB and MPS IVB; MPS IIIC and MPS IVB; MPS I and MPS IVB; MPS IX and MPS IIIA; MPS IX and MPS IIIB; MPS IX and MPS IIIC; MPS IX and MPS IIID; MPS IX and MPS IVB; MPS VII and MPS IIIB; MPS VII and MPS IVB; and MPS IIIA and MPS IIIB.

In cellular response to stimulus, the highest similarities were between the following: MPS IVB and MPS IIIA; MPS IVB and MPS IIIB; MPS IVB and MPS IIIC; MPS IVB and MPS IX; and MPS IX to all subtypes of MPS III.

In signal transduction, the highest similarities were between the following: MPS IVB and MPS IIIA; MPS IVB and MPS IIIB; MPS IVB and MPS IIIC; MPS IVB and MPS I; MPS IVB and MPS IX; MPS IX and MPS IIIA; MPS IX and MPS IIIB; MPS IX and MPS IIIC; and MPS VII and MPS IVB.

In positive regulation of cellular processes, the highest similarities were between the following: MPS IX and all subtypes of MPS III; MPS IX and MPS IVB; MPS IVB and MPS IIIA; MPS IVB and MPS IIIB; MPS IVB and MPS IIIC; and MPS IVB and MPS VII.

In negative regulation of cellular processes, the highest similarities were between the following: MPS IX and MPS IIIB; MPS IX and MPS IIIC; MPS IX and MPS IIID; MPS IX and MPS IVB; MPS IVB and MPS IIIA; MPS IVB and MPS IIIB; and MPS IVB and MPS IIIC.

Finally, in cellular developmental process, the highest similarities were between th following: MPS IX and MPS IIIB; MPS IX and MPS IIIC; MPS IX and MPS IVB; MPS IVB and MPS IIIC; MPS IVB and MPS IIIA; and MPS IIIB and MPS VII. These results indicate that there are clusters of MPS types in which similar changes in expression of genes coding for proteins involved in cellular processes are present. The most obvious cluster consists of MPS III (all subtypes), MPS IVB, and MPS IX. Then, MPS VII and MPS I reveal a high similarity to this cluster of MPS types.

To define genes which expressions are particularly affected in MPS, we have identified transcripts whose levels were changed especially strongly (FDR < 10^−6^, *p* < 0.01, log_2_ fold change (FC)> 2.5). The results are presented in Table 3 and Table 4. Interestingly, there are specific transcripts which levels are drastically changed in several MPS types. Genes whose expression is changed 5.6-times or more (log_2_ FC > 2.5) in at least six MPS types include *LY6K, COL8A2, CAPG, CLU, ADAMTSL1, POSTN,* and *MFAP5*, coding for lymphocyte antigen 6 family member K, alpha 2 chain of type VIII collagen, capping actin protein–gelsolin like, clusterin, ADAMTS-like 1 protein, periostin, and MFAP5 glycoprotein, respectively. Therefore, one may assume that regulation of cell growth, formation of endothelia, cytoskeleton formation, apoptosis, cell cycle regulation, DNA repair, and functions of the extracellular matrix and microfibrils can be significantly disturbed in MPS cells due to huge changes in expression of crucial genes (see Discussion). Again, this can be a basis for explaining the appearance of specific MPS symptoms in patients (compare [4] with Figure 2, Appendix A; see Discussion for specific analyses and more details).

In addition, we have analyzed in more detail mRNA levels of some other genes which appeared especially interesting. In these cases, the RNA-seq results were confirmed by RT-qPCR analyses (Figure 5). Apart from data for specific genes, results presented in Figure 2 have demonstrated that changes in RNA levels, reported in this paper, can be confirmed by two independent methods; thus, the values obtained in the global transcriptomic analysis (RNA-seq) can be considered reliable.

As shown in Figure 5, expression of the following genes is significantly changed relative to HDFa control cells: *CLU* in MPS II, *MFGE8* in MPS IIIA, *HIP1* in MPS IIIB, *COL5A1* in MPS IIIC, *STS* in MPS IIID, *SCARA3* in MPS VI, *MAN2A1* in MPS VII, and *ATF5* in MPS IX. These results indicate that dysregulation of expression of the abovementioned genes may contribute to the appearance of the following disturbances in cellular and tissue-related processes: apoptosis, cell cycle regulation, and DNA repair in MPS II (however, note that *CLU* expression is enhanced in several MPS types; Table 4); phagocytosis, angiogenesis, atherosclerosis, tissue remodeling, and hemostasis regulation in MPS IIIA; endocytosis and cellular trafficking in MPS IIIB; functions of the connective tissue in MPS IIIC; production of estrogens, androgens, and cholesterol in MPS IIID; oxidative stress response in MPS VI; oligosaccharide (N-glycan) maturation in MPS VII; and cell differentiation and cellular adaptation to stress in MPS IX (see Discussion for detailed description and analyses).

## 3. Discussion

MPS consists of a group of 11 genetic disorders in which degradation of GAG is impaired [4]. Each of MPS type/subtype is caused by dysfunction of a single gene which codes for an enzyme involved in GAG degradation (Table 1). Undegraded GAGs are stored in lysosomes, and this is considered to be the primary cause of the disease. Clinically, there are some common symptoms of MPS; however, characteristic symptoms for each MPS type are correlated with the kind of GAG(s) stored in lysosomes (Table 1). MPS I patients, in which heparan sulfate and dermatan sulfate are accumulated, suffer from short stature, dysmorphology, organomegaly, dysostosis multiplex, joint stiffness, dysfunctions of visceral organs and sensory organs (including corneal clouding, retinal degeneration, and hearing deficits), and severe learning and cognitive deficits, followed by progressive deterioration of the central nervous system functions in a subset of patients. MPS II patients, in which heparan sulfate and dermatan sulfate are also accumulated, have symptoms similar to MPS I, but corneal clouding is absent, and aggressive-like behavior occurs in patients with neuronopathic forms. MPS III patients, who accumulate heparan sulfate in lysosomes, have relatively milder somatic symptoms, while dysfunctions of the central nervous system are extremely severe, including hyperactivity, sleep disturbance, severe mental retardation, loss of communication skills, and aggressive-like behavior, followed by lack of mobility and fatal neurodegeneration. In MPS IV, keratan sulfate is stored, and no mental or cognitive dysfunctions occur, while severe skeletal disturbances are evident, leading to very short stature and skeletal-dysfunction-related disorders. In MPS VI, dermatan sulfate is stored, and similarly to MPS IV, no neurodegeneration occurs, while patients suffer from severe somatic symptoms which are, however, similar to those in MPS I and MPS II rather than MPS IV. MPS VII is characterized by the accumulation of heparan sulfate, dermatan sulfate, and chondroitin sulfate, and symptoms resembling those of MPS I and MPS II, but with usually normal intelligence. MPS IX patients accumulate hyaluronic acid and suffer from joint dysfunctions, short stature, and erosion of the hip joint, while intelligence is not affected. In light of the variability of symptoms of different MPS types, one might ask what is the reason of such differences in symptoms between MPS types? If the lysosomal GAG storage and resultant dysfunctions of these organelles were the only cause of the disease, symptoms of all MPS types should be similar, irrespective of the type of stored GAG(s). This is definitely not the case. This problem has been recognized previously (for extensive discussions, see [4,7]), but many attempts of various research groups to understand molecular mechanisms of differences between symptoms of various MPS types were as yet unsuccessful, and it is not possible to present a comprehensive model for such mechanisms. On the other hand, one could imagine that if undegraded GAGs are able to leak from damaged lysosomes, or even from damaged cells, they might affect various cellular processes differentially, depending on their kinds. In fact, it was hypothesized that different behaviors of patients suffering from various MPS types may arise from differences of chemical moieties exposed at the ends of partially degraded GAGs, depending on the stage of degradation which is blocked due to deficiency of particular lysosomal enzyme [17]. However, such a hypothesis can be true only if one assumes actions of large amounts of partially degraded GAGs, not only inside lysosomes but also in cytoplasm or outside cells. If so, one could speculate that such GAGs would affect different cellular processes directly rather than only through cellular stress caused by lysosomal dysfunction. Such a scenario would include changed expression of a battery of genes, causing amplification of disturbance of various cellular processes.

To test the above hypothesis, we performed transcriptomic studies, using the RNA-seq technique. Although a few transcriptomic studies in MPS were reported previously [9,18,19], this work is the first one in which all types of MPS were investigated simultaneously, using the same model (human fibroblasts, in this case). We consider this to be the major advantage of this study, which should prompt further investigations on global changes in genetic and cellular processes in MPS, in order to understand details of molecular mechanisms of this group of diseases. This may strongly facilitate development of novel therapeutic options which could be effective in management of these severe disorders. On the other hand, we are aware of some limitations of this study. First, only one cell line of each MPS type was investigated. This is due to both technical and economical restrictions which are connected to studies on all MPS types together. Four biological repeats (which are necessary to obtain reliable results) of experiments on 12 cell lines (11 MPS types and subtypes and a control) caused a serious challenge, both technical and financial, which, at this stage of research, would be very difficult to overcome if several or many cell lines from each MPS type were tested. However, any interpretation of the results of experiments with only one cell line from each MPS type must take into consideration genetic variability of patients within every single disease. Therefore, finding common changes in expression of genes in many MPS types relative to the control cell line (as demonstrated in this report) can be an indication that obtained results are disease-specific rather than representing random fluctuations. Second, as in all experiments with cell cultures, differences in results between one and another biological repeat can be considerable, as even minor variations in culturing conditions or the age (number of passages) of the cell line might influence transcriptomes. This is why four biological repeats for each cell line were necessary and detailed statistical analyses (according to commonly accepted procedures) had to be conducted to make any solid conclusions. Third, the use of fibroblast was convenient in in vitro studies, as such cells could be obtained from various MPS patients by using a mildly invasive method, and it allowed researchers to standardize conditions for all MPS types and controls. However, one should note that various genes may be differentially expressed in different cell types. Moreover, the most severe MPS symptoms do not involve pathology of fibroblasts. Therefore, to obtain a complete picture of transcriptomes of all MPS types, ideally, cells from different tissues should be investigated. Nevertheless, since this appears rather unrealistic at the moment, we considered that performing transcriptomic studies using fibroblast lines derived from patients suffering from all MPS types and subtypes may be an important starting point to understand global molecular changes in these diseases and to stimulate a new way of research that facilitates the creation of a comprehensive model of MPS pathomechanisms.

We found global changes in gene expression patterns in each MPS type relative to wild-type cells, ranging from 289 to 893, significantly changed transcripts (Figure 1). This very global result indicated significant variabilities between different MPS types. On the other hand, when analyzing transcripts of significantly changed levels in most MPS types, we could identify processes with particularly high number of miss-regulated genes in MPS relative to control (HDFa) fibroblasts. The processes grouping a large proportion of up- and/or down-regulated genes coding for proteins involved in them include protein folding, vesicle targeting, microtubule-based processes, actin filament-based processes, cell cycle, cell communication, signal transduction, positive and negative regulation, cell growth, cell development, cell division, cellular homeostasis, autophagy, and cell death (Figure 2, Appendix A). Since the changes were quite common for most MPS types, one might assume that miss-regulation of corresponding genes contributes to the development of symptoms common for the whole group of GAG storage diseases. For example, disturbance of cell growth, cell cycle, and cell development might contribute to the short stature of patients, and microtubule-based processes and actin filament-based processes may result in muscle problems, while dysregulation of protein folding, vesicle targeting, cellular homeostasis, autophagy, and cell death control can be likely involved in multiple dysfunctions of visceral organs and the central nervous system.

On the other hand, when considering processes potentially changed to the highest extent (with at least 30 kinds of transcripts particularly strongly changed), we found both common and specific features of particular MPS types (Figure 3). Huge changes in expression of genes involved in cellular metabolic processes occurred in all MPS types, indicating that severe changes in global regulation of metabolism are a general feature of MPS. Perhaps they contribute significantly to the development of common MPS symptoms. Disturbances in cell communication, cellular component organization and cellular response to stimulus can be predicted (on the basis of particularly severe changes in expressions of corresponding genes) in all types but MPS IVA, while those in signal transduction occur in all types but MPS IVA and MPS VI. These two types of MPS are characterized by significant changes in somatic tissues but normal development and functions of central nervous system, but these features are ascribed to the lack of keratan sulfate and dermatan sulfate in neurons. On the other hand, relatively milder disturbances of the regulatory processes, which are listed above, in MPS IVA and MPS VI might facilitate proper functioning of neurons if surrounding tissues are also affected only mildly. This suggestion might be corroborated by the observation that particularly deep changes in positive and negative regulations could be predicted in all types, except MPS II, MPS IVA, and MPS VI. Type II of MPS is clinically quite similar to MPS I, while the latter type is considered more severe if no residual activity of the appropriate enzyme is present [4]. Therefore, disruption of regulatory pathways might result in particularly severe MPS symptoms.

When we compared number of genes coding for proteins involved in particular processes which expression was changed and which were common to different MPS types, we found some intriguing correlations. Namely, when considering nine different processes, it was possible to identify clusters of MPS types revealing high similarity in patterns of expression of many genes. The most evident cluster contains all subtypes of MPS III, MPS IVB, and MPS IX. Significant similarity to this cluster can be also found in MPS VII and MPS I (Figure 4). Thus, one might assume that cellular changes can be quite similar in all these MPS types. Intriguingly, different kinds of GAGs are stored in all these MPS types (see Table 1), suggesting that there is no direct correlation between the nature of GAGs and cellular disturbances occurring in cells accumulating these compounds. On the other hand, previously reported analyses suggested that not the kind of GAG per se, but the stage at which its degradation is stopped, resulting in exposition of specific chemical moiety at the end of incompletely degraded macromolecule, might be responsible for specific effects in organisms of MPS patients [17].

Generally, although previous reports signaled some specific changes in cellular structures and processes in MPS, they included only lysosomes, cytoskeleton, vacuolar transport, cell cycle mitochondrial enzymes, apoptosis, and autophagy [9,10,11,12,13,14,15,16]. Results presented in this report indicate that this list should be significantly enlarged. Moreover, comparison of all MPS types provides a basis for global assessment of changes occurring in each particular type, as well as in every type or most of types.

When analyzing particular genes for which expression is extremely affected (above 5.6 times, i.e., log_2_ FC > 2.5) in at least two MPS types (Table 4), some very interesting findings are especially worth discussing in detail. Levels of *LY6K* gene transcripts are severely decreased in all MPS types, except MPS IIIA, IIIB, and IVA. This gene codes for lymphocyte antigen 6 family member K, which may be involved in the regulation of cell growth, as it was demonstrated that silencing of *LY6K* expression by siRNA resulted in growth suppression of lung and esophagus cancer cells [20]. Therefore, one can assume that cell growth inhibition due to down-regulation of *LY6K* expression in MPS cells can contribute to developmental problems, resulting in the short stature of patients and functional deficits of their organs.

Expression of *COL8A2,* coding for alpha 2 chain of type VIII collagen, is highly up-regulated in seven MPS types. This protein is a major component of corneal endothelial cells and the endothelia of blood vessels [21]. Hence, it is possible that highly elevated levels of this protein may be involved in cornea clouding and/or blood-vessel dysfunctions. Intriguingly, cornea clouding occurs in MPS I and MPS VI, but is absent in MPS II [2], and overexpression of *COL8A2* has also been detected in MPS I and MPS VI, but not in MPS II (Table 4).

The *CAPG* gene, coding for capping actin protein–gelsolin like, is up-regulated in six MPS types. The product of this gene modulates functions of actin, thus playing a role in cytoskeleton formation [22]. One may predict that cytoskeleton changes can significantly influence cell functions, leading to various dysfunctions of tissues and organs.

Clusterin, encoded by the *CLU* gene, is an extracellular chaperone that prevents aggregation of non-native proteins. Moreover, it was demonstrated that clusterin is involved in various cellular processes, including apoptosis, cell cycle regulation, and DNA repair [23]. Overexpression of the *CLU* gene in six MPS types suggests that this might be a response to stress conditions caused by cell dysfunction. Importantly, *CLU*-derived transcript level was previously reported to be increased nine times in tissues of aortas isolated from MPS I dogs and mice, relative to control animals [24], and corresponds closely to levels of *CLU* overexpression in fibroblasts from various MPS types measured in this study.

The *ADAMTSL1* gene codes for ADAMTS-like 1 protein. This protein may have important functions in the extracellular matrix [25]. Particularly, it cleaves aggrecan, the cartilage-specific proteoglycan or chondroitin sulfate proteoglycan 1 [26]. Therefore, overexpression of the *ADAMTSL1* gene in six MPS types may indicate a response of the cell to accumulation of GAGs not only in lysosomes, but also outside the cell. Similar explanation can be proposed for overexpression of the *POSTN* gene, coding for periostin, an extracellular matrix protein that functions in tissue development and regeneration [27], as well as for overexpression of the *MFAP5* gene, encoding a glycoprotein which is a component of microfibrils of the extracellular matrix [28].

Some genes are expressed in an extremely changed mode in only a few MPS types; however, they deserve particular attention. For example, *HOXB5* and *HOXB6* code for proteins belonging to the Antp homeobox family and encode nuclear proteins with homeobox DNA-binding domains, participating in development of lung and gut (*HOXB5*), and lung and skin (*HOXB6*), respectively [29,30]. *NOTCH3* encodes Notch Receptor 3, which is involved in various developmental processes, including vasculogenesis neural development [31]. Expression of the abovementioned genes is significantly changed in some MPS types (Table 4), and changes in lung, gut, skin, vascular system, and central nervous system are evident in MPS patients. If expression of these genes is affected in MPS patients during embryonic development, one might speculate that dysfunctions of these organs may partially arise from dysregulation of development-associated genes, even before clinical symptoms are evident. Interestingly, significant changes in expression of many genes involved in development have been reported previously in MPS VII mice [32].

There is also a group of genes in which expression is changed only in particular MPS types, but which appear particularly interesting in the light of the disease mechanisms. *MFGE8* is highly overexpressed in MPS IIIA (Table 4 and Figure 5). It encodes lactadherin, a multifunctional protein involved in phagocytosis, angiogenesis, atherosclerosis, tissue remodeling, and hemostasis regulation [33]. Therefore, these crucial cellular and tissue processes are likely dysregulated in MPS IIIA patients.

The *HIP1* gene codes for Huntingtin Interacting Protein 1, playing a role in clathrin-mediated endocytosis and trafficking [34]. Expression of *HIP1* is enhanced significantly in MPS IIIB (Figure 5), indicating that endocytosis is stimulated and cellular trafficking is affected. These effects may likely contribute to specific symptoms of MPS IIIB, as in the case of Huntington’s disease, in which *HIP1* gene product cannot properly interact with mutant Huntingtin variant [34].

Collagen type V alpha 1 chain is encoded by the *COL5A1* gene, significantly overexpressed in MPS IIIC. This collagen type plays a significant role in functions of the connective tissue [35], and, thus, abnormal expression of *COL5A1* may contribute to impairment of these functions.

The steroid sulfatase is the *STS* gene product. This enzyme catalyzes the hydrolysis of various 3-beta-hydroxysteroid sulfates, which are precursors for crucial biologically active compounds such as estrogens, androgens, and cholesterol [36]. Expression of *STS* is considerably impaired in MPS IIID (Figure 5). Hence, it is likely that some hormonal and cellular (related to cholesterol roles) functions are impaired in MPS IIID patients due to decreased levels of the steroid sulfatase enzyme.

*SCARA3* codes for scavenger receptor class A member 3. This protein is involved in depletion of reactive oxygen species and, thus, in the protection of cells from oxidative stress [37]. Expression of *SCARA3* is known to be induced under conditions of oxidative stress; thus, significantly enhanced levels of *SCARA3* mRNA in MPS VI cells indicated that this kind of stress is severe in these cells, with all the consequences for cellular processes.

Expression of the *MAN2A1* gene, coding for mannosidase alpha class 2A member 1, is impaired in MPS VII (Figure 5). Since this enzyme, localized in the Golgi apparatus, is involved in the asparagine-linked oligosaccharide (N-glycan) maturation pathway [38], its lowered amount may contribute to secondary carbohydrate metabolic defects in MPS VII.

*ATF5* encodes activating transcription factor 5 [39]. This protein is involved in the regulation of expression of various genes related to cell differentiation and cellular adaptation to stress [39]. Therefore, up-regulation of *ATF5* expression in MPS IX (Figure 5) indicates considerable stress conditions in the cells and suggests defects in cellular differentiation in this MPS type.

Such analyses of particular genes in which expressions are severely changed in MPS cells provide the starting point for more detailed studies, in the future, on specific changes in cellular processes occurring in various MPS types. Definitely, the disturbance of many cellular processes contributes significantly to pathomechanisms of MPS, and this field has not been investigated sufficiently to date. One should ask whether huge changes in expression of hundreds of genes are only tertiary or quaternary effects of GAG storage, due to lysosomal dysfunctions leading to disruption of cell metabolism and resultant stress responses and global perturbations in cell physiology, or if they are causes of various cellular dysfunctions. If the former hypothesis was true, one should expect similar patterns of gene expression changes in all MPS types, as lysosomal dysfunctions should be very similar, irrespective of the kind(s) of stored GAG(s). This is, however, not the case, as differences between gene expression patterns in various MPS types are very considerable. On the other hand, results of RNA-seq analyses (supported by RT-qPCR of selected genes) presented in this report might be explained if primary GAG storage caused partial disruption of lysosomes, leakage of different kinds of GAG(s) in different MPS types, and, thus, different interactions of GAGs with cellular structures and proteins, perhaps including transcription factors. Further studies are required to test both hypotheses; nevertheless, we suggest that the latter one is more likely.

## 4. Materials and Methods

### 4.1. Cell Lines

Fibroblasts (cell lines) derived from patients suffering from MPS types I, II, IIIA, IIIB, IIIC, IIID, IVA, IVB, VI, VII, and IX (Table 2), as well as control HDFa cell line, were used. These cell lines were obtained from the NIGMS Human Genetic Cell Repository at the Coriell Institute for Medical Research (ID numbers: GM00798, GM13203, GM00879, GM00156, GM05157, GM05093, GM00593, GM03251, GM03722, GM00121, and GM17494). They were cultured in vitro, using DMEM medium supplemented with antibiotics and 10% fetal bovine serum (FBS) under standard conditions (37 °C, 95% humidity, and atmosphere saturated with 5% CO_2_).

### 4.2. RNA Isolation and Purification

First, 5 × 10^5^ cells were passaged on plates (10 cm diameter) and allowed to attach overnight. Cells were lysed with a solution containing guanidine isothiocyanate and β-mercaptoethanol, to effectively inactivate RNAses, and homogenized by using the QIAshredder column, followed by RNA extraction with the RNeasy Mini kit (Qiagen, Hilden, Germany) and treatment with Turbo DNase (Life Technologies, Life Technologies, Carlsbad, CA, USA), according to the manufacturers’ instructions. The quality of isolated RNA samples was assessed by RNA Nano Chips (Agilent Technologies, Santa Clara, CA, USA) in Agilent 2100 Bioanalyzer System.

### 4.3. RNA-Seq Analysis

The mRNA libraries were generated with Illumina TruSeq Stranded mRNA Library Prep Kit. The cDNA libraries were sequenced on a HiSeq4000 (Illumina, San Diego, CA, USA), with the following parameters: PE150 (150bp paired-end) and minimum 40 million of raw reads (40M), which gave a minimum of 12 Gb of raw data per each sample. Quality assessment was carried out by FastQC version v0.11.7. The RNA-seq data were submitted to the NCBI Sequence Read Archive (SRA): PRJNA562649. Raw readings were mapped to the GRCh38 human reference genome from the Ensembl database, using the Hisat2 v. 2.1.0 program. To calculate the expression level of the transcripts, the Cuffquant and Cuffmerge programs in version 2.2.1 and the GTF Homo_sapiens.GRCh38.94.gtf file from the Ensembl database (https://www.ensembl.org/index.html as of 19 February, 2019) were used. The Cuffmerge program was started with the library-norm-method classic-fpkm parameter, normalizing the expression values by means of the FPKM algorithm. Statistical significance was analyzed by using one-way analysis of variance (ANOVA) on log_2_(1 + x) values which have normal continuous distribution [14]. The false discovery rate (FDR) was estimated by using the Benjamini–Hochberg method. For comparisons between two groups, post hoc Student’s *t*-test with Bonferroni correction was employed. All statistical analyses were performed by using R software v3.4.3. Transcript annotation and classification was performed by using the BioMart interface for the Ensembl gene database (https://www.ensembl.org/info/data/biomart/index.html, as for 19 February, 2019).

### 4.4. RT-qPCR Analysis

Reverse transcription–quantitative real-time PCR (RT-qPCR) was performed in order to measure the mRNA levels of the selected genes. Total RNA was subjected to reverse transcription, using iScript Reverse Transcription Supermix for RT-qPCR (BioRad, Hercules, CA, USA), according to the manufacturer’s instructions. RT-qPCR was carried out on PrimePCR^TM^ Custom Plates, using SsoAdvanced^TM^ Universal IT SYBR Green SMx (BioRad, USA) with Unique Assay ID as follows: qHsaCID0037852 for *ATG5*, qHsaCID0012475 for *CLU*, qHsaCID0014514 for *COL5A1*, qHsaCID0015337 for *MFGE8*, qHsaCID0017812 for *MAN2A1*, qHsaCID0006298 for *HIP1*, qHsaCID0008300 for *SCARA3*, and qHsaCID0015255 for *STS* with normalization relative to two housekeeping genes *GAPDH* (qHsaCED0038674) and *G6PD* (qHsaCED0001353), using CFX96 Touch Real-Time PCR Detection System (BioRad). Estimation of gene expression was performed by the 2^−ΔΔC(T)^ method. One-way ANOVA (*p* < 0.05) was used to test statistical-significance differences between patients’ derived fibroblasts and control cells.

## 5. Conclusions

Results of global transcriptomic analyses in all types of MPS indicated that there are multiple and significant changes in levels of expression of many genes whose products are involved in various cellular processes. Therefore, not only dysfunctions of cells caused by physical storage of GAG in lysosomes, but also disturbances in cellular processes, caused by dysregulation of different genes are responsible for cellular disorders observed in MPS. Assumptions on importance in secondary and tertiary cellular changes were reported previously (as summarized and discussed recently [40]); however, they were not linked to global changes in genes’ expression, and all types of MPS were not investigated in this aspect previously. Our results point to possible molecular mechanisms of both common features of MPS and variability between particular MPS types, indicating the way to study this problem in more detail, to find precise causes of changed gene expression and disturbances of cellular processes. This may also lead to the development of novel therapeutic strategies. Currently available therapies fail to correct all MPS symptoms, and despite their known limitations, like inability of the recombinant enzyme to cross the blood–brain barrier, as well as to reach bones effectively, one may speculate that this is partially due to the appearance of disorders of cellular processes, resulting indirectly from GAG storage, but advanced to such a stage that subsequent reduction of GAG accumulation cannot reverse them. Therefore, one can suggest that only combined therapies, which should involve drugs correcting specific cellular processes, along with reducing GAG storage, may lead to the obtainment of a real cure for MPS. Finally, this report demonstrates that pathomechanisms of diseases considered as monogenic, like MPS, may actually involve dysregulation of a battery of genes, being in fact “multigenic”.

## Figures and Tables

**Figure 1 ijms-21-01204-f001:**
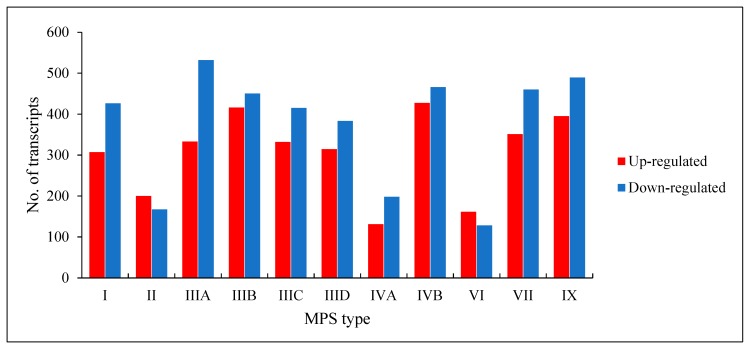
Number of up- and down-regulated transcripts (at FDR < 0.1; *p* < 0.1) in different types of MPS relative to control cells (HDFa).

**Figure 2 ijms-21-01204-f002:**
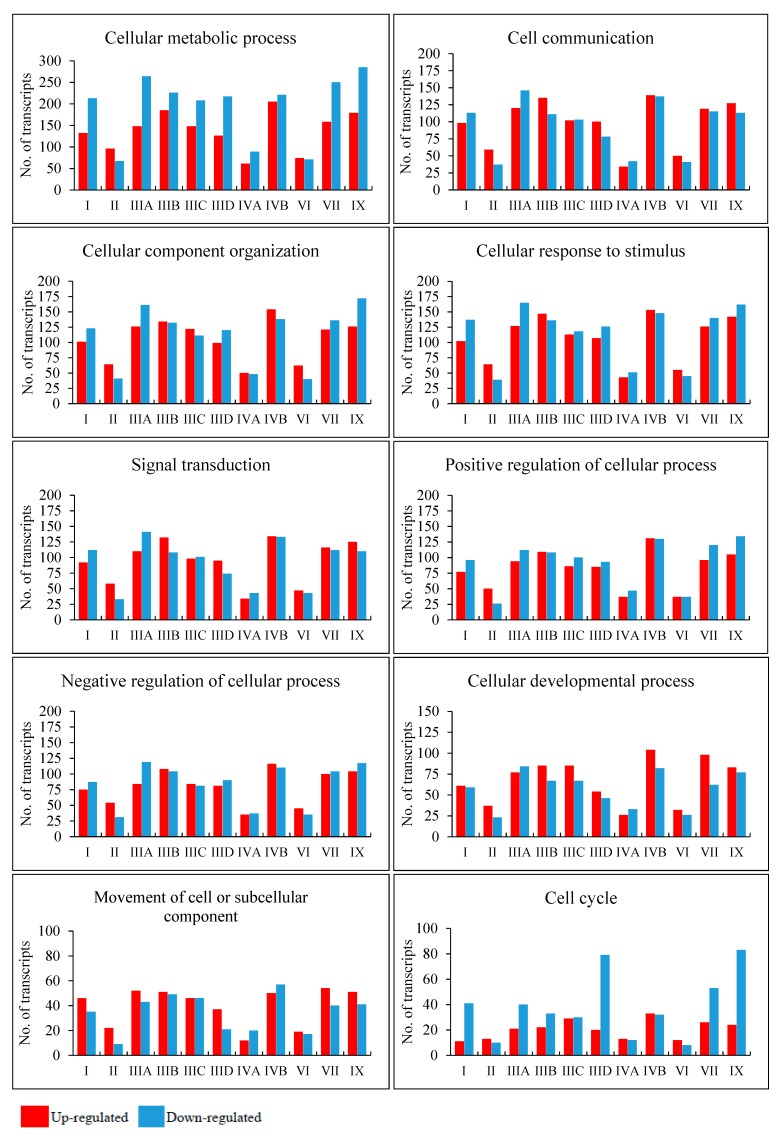
Number of up- and down-regulated transcripts (at FDR < 0.1; *p* < 0.1) with division into selected cellular processes in different MPS types relative to control cells (HDFa).

**Figure 3 ijms-21-01204-f003:**
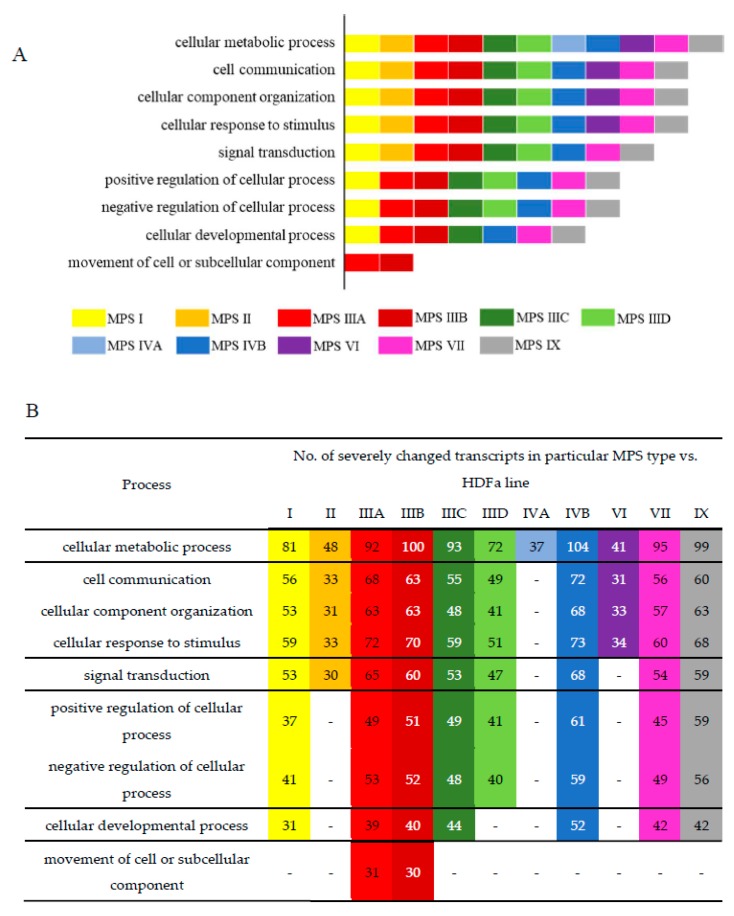
Cellular processes in which at least 30 kinds of transcripts were particularly strongly changed (FDR < 10^−6^, *p* < 0.1) in different MPS types. The “rainbow” diagram (**A**) shows these processes in order to demonstrate which process in significantly affected in which MPS types. The colors correspond to MPS types shown in panel (**B**), in which the number of severely changed transcripts of genes which products are involved in particular processes are indicated.

**Figure 4 ijms-21-01204-f004:**
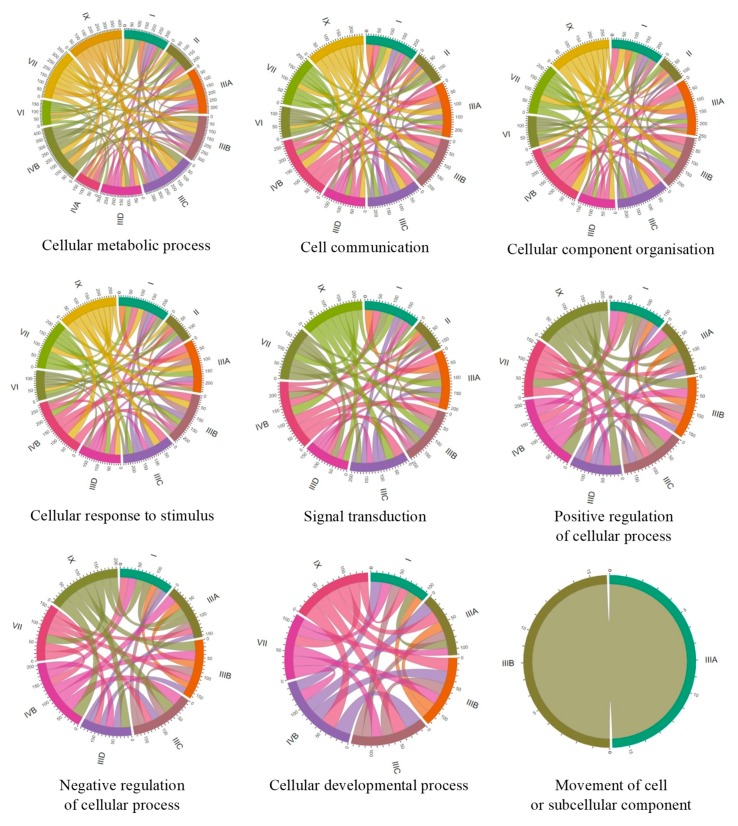
Chord diagrams indicating numbers of common genes between individual MPS types for which expression is particularly strongly changed (FDR < 10^−6^, *p* < 0.1) relative to control cell (HDFa line). Individual panels group genes involved in indicated cellular processes. Roman numbers indicate corresponding MPS types.

**Figure 5 ijms-21-01204-f005:**
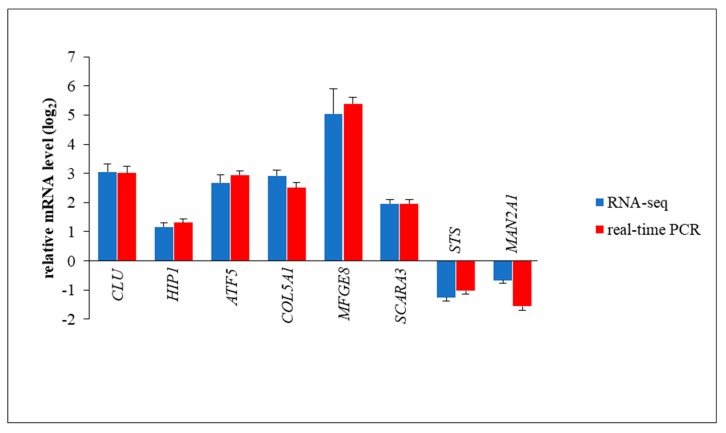
Comparison of results of RNA-seq (blue columns) and RT-qPCR (red columns) for selected genes in various MPS types relative to wild-type HDFa cells. Results for HDFa are normalized as a horizontal line (value 0), and values for particular MPS types reflect this value, with changes in transcript level indicated as log_2_ fold change (FC). Results are presented as mean values from four biological experiments, with error bars indicating SD. In all cases, the differences between MPS cells and HDFa cells were statistically significant (*p* < 0.05 in one-way ANOVA). Particular genes were tested in different MPS types, as follows: *CLU* in MPS II, *HIP1* in MPS IIIB, *ATF5* in MPS IX, *COL5A1* in MPS IIIC, *MFGE8* in MPS IIIA, *SCARA3* in MPS VI, *STS* in MPS IIID, and *MAN2A* in MPS VII.

**Table 1 ijms-21-01204-t001:** Types and subtypes of MPS (based on [4]).

MPS Type	OMIM No. (Gene-Locus MIM/PMIM)	Defective Enzyme (EC No.)	Stored GAG(s)
MPS I	252800/607014, 607015, 607016	α-l-iduronidase (EC 3.2.1.76)	Heparan sulfate, Dermatan dulfate
MPS II	300823/309900	2-iduronate sulfatase (EC 3.1.6.13)	Heparan sulfate, Dermatan dulfate
MPS IIIA	605270/252900	*N*-sulfoglucosamine sulfhydrolase (EC 3.10.1.1)	Heparan sulfate
MPS IIIB	609701/252920	α-*N*-acetylglucosaminidase (EC 3.2.1.50)	Heparan sulfate
MPS IIIC	610453/252930	Acetyl-CoA:α-glucosaminide acetyltransferase (EC 2.3.1.78)	Heparan sulfate
MPS IIID	607664/252940	*N*-acetylglucosamine 6-sulfatase (EC 3.1.6.14)	Heparan sulfate
MPS IVA	612222/253000	*N*-acetylgalactosamine 6-sulfatase (EC 3.1.6.4)	Keratan sulfate, Chondroitin sulfate
MPS IVB	611458/253010	β-galactosidase-1 (EC 3.2.1.23)	Keratan sulfate
MPS VI	611542/253200	*N*-acetylgalactosamine 4-sulfatase (EC 3.1.6.12)	Dermatan sulfate
MPS VII	611499/253220	β-glucuronidase (EC 3.2.1.31)	Heparan sulfate, Dermatan sulfate, Chondroitin sulfate
MPS IX	607071/601492	Hyaluronidase-1 (EC 3.2.1.35)	Hyaluronic acid

**Table 2 ijms-21-01204-t002:** Characteristics of MPS patients’ derived fibroblast used in this study.

Cell Line	Race *	Sex *	Age *#	Mutated Gene and Its Locus *	Mutation *			
					Allele 1		Allele 2	
					DNA	Protein	DNA	Protein
MPS I	Caucasian	Female	1	*IDUA*, 4p16.3	G1293A	Trp402X	G1293A	Trp402X
MPS II	Caucasian (ethnicity: Haitian)	Male	3	*IDS*, Xp28	208insC	His70ProfsX29	none	none
MPS IIIA	Caucasian	Female	3	*SGSH*, 17q25.3	G1351A	Glu447Lys	G746A	Arg245His
MPS IIIB	Caucasian	Male	7	*NAGLU*, 17q21	C1876T	Arg626X	C1876T	Arg626X
MPS IIIC	unknown	Male	8	*HGSNAT*, 8p11.1	ND	ND	ND	ND
MPS IIID	Asian Indian	Male	7	*GNS*, 12q14	C1063T	Arg355X	C1063T	Arg355X
MPS IVA	Caucasian (ethnicity: Mexican)	Female	7	*GALNS*, 16q24.3	ND	ND	ND	ND
MPS IVB	Caucasian	Female	4	*GLB1*, 3p22.3	TG851-852CT	Trp273Leu	G1561T	Trp509Cys
MPS VI	Black	Female	3	*ARSB*, 4q14.1	ND	ND	ND	ND
MPS VII	African American	Male	3	*GUSB*, 7q21.11	G1881T	Trp627Cys	G1068A	Arg356X
MPS IX	unknown	Female	14	*HYAL1*, 3p.21.3	ND	ND	ND	ND

* According to cell line description in Coriell Institute; #age (years) at the time of cell collection; ND = not determined.

**Table 3 ijms-21-01204-t003:** Number of up- and down-regulated genes with log_2_ FC > 2.5 in different types of MPS relative to control cells (HDFa).

Genes	No. of Genes with log_2_ FC > 2.5 in Particular MPS Type vs. HDFa Line										
	I	II	IIIA	IIIB	IIIC	IIID	IVA	IVB	VI	VII	IX
Up-regulated	6	5	11	18	16	6	2	15	5	13	14
Down-regulated	5	3	6	10	9	6	0	16	2	13	9
**Total**	**11**	**8**	**17**	**28**	**25**	**12**	**2**	**31**	**7**	**26**	**23**

**Table 4 ijms-21-01204-t004:** Selected up-regulated (bold **X**) and down-regulated (italic *X*) genes, with log_2_ FC > 2.5 in different types of MPS relative to control cells (HDFa), which are characterized with severe changes in expression in at least two different types of MPS. Detailed statistical analysis is shown in Appendix A.

Gene	Transcripts with log_2_ FC > 2.5 in Particular MPS Type vs. HDFa Line
I	II	IIIA	IIIB	IIIC	IIID	IVA	IVB	VI	VII	IX
*COL8A2*	**X**		**X**		**X**	**X**	**X**		**X**		**X**
*CAPG*	**X**				**X**		**X**	**X**	**X**	**X**	
*CLU*	**X**	**X**			**X**			**X**	**X**		**X**
*ADAMTSL1*	**X**		**X**	**X**	**X**				**X**		**X**
*POSTN*			**X**	**X**	**X**	**X**		**X**			**X**
*MFAP5*			**X**	**X**	**X**			**X**		**X**	**X**
*PCOLCE2*				**X**		**X**		**X**		**X**	**X**
*MFGE8*	**X**		**X**	**X**				**X**			
*FAM167A*			**X**	**X**	**X**			**X**			
*NR2F2*				**X**	**X**			**X**			**X**
*CDH2*				**X**	**X**			**X**			**X**
*TENM3*					**X**			**X**		**X**	**X**
*MN1*	**X**	**X**									**X**
*PFN1*		**X**	**X**		**X**						
*OXTR*				**X**	**X**			**X**			
*MT1X*				**X**				**X**			
*AC004556.1*				**X**		**X**					
*NOTCH3*										**X**	**X**
*LY6K*	*X*	*X*			*X*	*X*		*X*	*X*	*X*	*X*
*SERPINB7*		*X*				*X*		*X*	*X*		*X*
*ENPP2*			*X*	*X*	*X*			*X*			*X*
*CLEC2B*		*X*	*X*					*X*		*X*	
*SERPINB2*	*X*							*X*			
*SNHG5*	*X*			*X*						*X*	
*CTSC*			*X*							*X*	
*PTGDS*			*X*	*X*							
*PTGS1*			*X*							*X*	
*COL18A1*			*X*								*X*
*TRPV2*				*X*							*X*
*KREMEN1*				*X*				*X*			
*WISP2*				*X*				*X*			
*TNFRSF11B*				*X*						*X*	
*EPDR1*					*X*						*X*
*HOXB6*					*X*			*X*			
*HOXB5*					*X*			*X*			
*RPL10*						*X*				*X*

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
