# Peer review of "Underestimated Aspect of Mucopolysaccharidosis Pathogenesis: Global Changes in Cellular Processes Revealed by Transcriptomic Studies"

_ijms, 2020, doi:10.3390/ijms21041204_

Round 1
Reviewer 1 Report
Gaffke and co-workers present an extensive transcriptomic study on MPS diseases, providing original data for every single MPS disease and nicely discussing them. This is an actual tour-de-force, where each cell line (from the 11 different MPS diseases + a control) was analysed four times independently, with RNA-seq results supported by qRT-PCR. The overall results revealed surprisingly large changes in the expression of a number of genes involved in various cellular processes. Furthermore, there were also significant differences between the transcriptomic profiles of different MPS, which may help shed some light on the clinical differences observed between them (even when the same GAGs are stored).
To the best of my knowledge, this is the first time in which all MPS types and subtypes are investigated simultaneously using the same model and I definitely believe that the overall results from this study will prompt additional analysis and studies from other teams, which may bring some fundamental highlights on the pathophysiological cascade that underlies MPS.
There are just a few minor points I would like to see corrected and/or addressed:
On the introduction section, either in the main text or in table 1, the authors should add the OMIM reference# for each MPS disease; I also believe it would be nice if they could add the Enzyme Commission (E.C.) number for enzymes on the defective enzyme column. On the results section (page 8), figure 1A needs to be totally reformulated. In fact, it does not correlate with the text description neither with the table provided in 1B. I mean, the text (lines 183 to 187) totally reflect what we see in the table (figure 1B) but are completely opposite to what we see in figure 1A. I’ll give one example, but it applies to all MPS, since the circles are designed it such a way that the ‘common’ processes/pathways are the ones on the basis of the figure (where all the circles overlay) and those which are specific to the fewer number of diseases are on top. From the text+table we learn that changes in transcripts involved in cell communication, cellular component organization and cellular response to stimulus occurred in all types but MPS VI A and MPS VI. Instead, what we ‘see’ in the figures is that those transcripts were altered ONLY in MPS IVA and VI. That should be altered. Still on the results section (page9), when the authors state that they analysed a few genes in more detail (lines 214 to 216), they say “In these cases, as well as in the case of expression of the CLU gene in MPS II, (…)”. In that context, the reference to the CLU gene makes no sense at all and should be skipped. Instead, it is missing afterwards, on lines 220 to 222, where the authors list all the genes and cell lines in which they were investigates. On the discussion section the authors state that GAGs storage is considered to be the main “or even only” cause of the disease (page 11, line 248; but also page 12, line 271). I feel this is quite an abusive statement because the time when we thought storage was the sole trigger for pathology in LSD is now long gone. We may not know the mechanisms underlying it, but over the last decades a number of authors have been drawing attention to that fact, and whole groups have been making efforts to understand the ‘bigger picture’. I am not suggesting the authors should skip those sentences; just keep then ‘lighter’. Still on the discussion section (lines 289 and 295), the authors use the following terminology: “all/most MPS”. For me, it is not clear whether the authors want to keep both options (all or most) but, since the changes have been quantified and largely documented, I feel they should just chose the term that applies better for each circumstance. Finally, there seems to be a problem with page numbering. Please check it and correct.
Author Response
Reviewer #1:
We thank the reviewer for constructive comments. Changes in the manuscript, introduced in response to the review, are marked in red.
REVIEWER’S COMMENT: On the introduction section, either in the main text or in table 1, the authors should add the OMIM reference# for each MPS disease;
RESPONSE: The OMIM no. (both gene/locus MIM and PMIM numbers) were introduced into Table 1.
REVIEWER’S COMMENT: I also believe it would be nice if they could add the Enzyme Commission (E.C.) number for enzymes on the defective enzyme column.
RESPONSE: I suggested, EC numbers were introduced into Table 1.
REVIEWER’S COMMENT: On the results section (page 8), figure 1A needs to be totally reformulated. In fact, it does not correlate with the text description neither with the table provided in 1B. I mean, the text (lines 183 to 187) totally reflect what we see in the table (figure 1B) but are completely opposite to what we see in figure 1A. I’ll give one example, but it applies to all MPS, since the circles are designed it such a way that the ‘common’ processes/pathways are the ones on the basis of the figure (where all the circles overlay) and those which are specific to the fewer number of diseases are on top. From the text+table we learn that changes in transcripts involved in cell communication, cellular component organization and cellular response to stimulus occurred in all types but MPS VI A and MPS VI. Instead, what we ‘see’ in the figures is that those transcripts were altered ONLY in MPS IVA and VI. That should be altered.
RESPONSE: We agree that Figure 1A might be misleading. Therefore, we have replaced it with a new version which is easy to follow. Moreover, we have presented the results in the form of Chord graph, as a new Figure 4, and described the analysis in the text (lines 206-274).
REVIEWER’S COMMENT: Still on the results section (page9), when the authors state that they analysed a few genes in more detail (lines 214 to 216), they say “In these cases, as well as in the case of expression of the CLU gene in MPS II, (…)”. In that context, the reference to the CLU gene makes no sense at all and should be skipped. Instead, it is missing afterwards, on lines 220 to 222, where the authors list all the genes and cell lines in which they were investigates.
RESPONSE: We have changed the text according to reviewer’s recommendations. The text “as well as in the case of expression of the CLU gene in MPS II” has been deleted, and the results with the CLU gene were added where we list all genes tested in the RT qPCR analysis (lines 297, 300-301).
REVIEWER’S COMMENT: On the discussion section the authors state that GAGs storage is considered to be the main “or even only” cause of the disease (page 11, line 248; but also page 12, line 271). I feel this is quite an abusive statement because the time when we thought storage was the sole trigger for pathology in LSD is now long gone. We may not know the mechanisms underlying it, but over the last decades a number of authors have been drawing attention to that fact, and whole groups have been making efforts to understand the ‘bigger picture’. I am not suggesting the authors should skip those sentences; just keep then ‘lighter’.
RESPONSE: We have changed the text according to reviewer’s recommendations. It is now clearly stated that importance of secondary and tertiary changes in MPS was noted by many authors previously, though no comprehensive molecular model could be proposed to date (lines 57-60, 350-354, 550).
REVIEWER’S COMMENT: Still on the discussion section (lines 289 and 295), the authors use the following terminology: “all/most MPS”. For me, it is not clear whether the authors want to keep both options (all or most) but, since the changes have been quantified and largely documented, I feel they should just chose the term that applies better for each circumstance.
RESPONSE: As suggested by the reviewer, we have chosen “most MPS” in the text, as it better reflects the results presented in this article.
REVIEWER’S COMMENT: Finally, there seems to be a problem with page numbering. Please check it and correct.
RESPONSE: Page numbering has been corrected.
Reviewer 2 Report
Article review
Underestimated aspect of MPS pathogenesis: global changes in cellular processes revealed by transcriptomic studies – Int. J. Mol. Sci.
In this manuscript, Gaffke et al. report interesting results of RNA-seq analysis using fibroblasts issued from MPS patients, pointing to the disruption of multiple cellular processes and furthering the understanding of pathogenesis in those LSDs as well as explaining the shortcomings of currently available treatments. They show up- and down-regulation of the expression of multiple genes involved in various cellular processes, some common to all 11 MPS, others specific to one or several subtypes. The authors confirmed their most significant results by RT-qPCR. They suggest that the underlying pathogenesis of mucopolysaccharidoses is not only linked to lysosomal storage, but also to the type of products that accumulate which may impact secondarily specific cell pathways, explaining the differences between MPS subtypes.
This manuscript is a relevant as monogenic diseases are redefined as complex disorders with the help of omics approaches. However, many limitations and errors weaken the message of the manuscript. Several tables and figures are difficult to understand and are not easy to read. The authors did not discuss the limitations of the study. Extensive English editing will enhance the scientific impact of this manuscript.
Major points:
The authors stated that they used four biological repeats for the experiments, but they do not detail how they obtained these biological replicates. Besides, studying only one sample per pathology represents a serious limitation that has not been discussed. In addition, there are inconstancies in gene expression in the biological repeats and this point is not discussed. Are the fibroblasts derived from cell lines or from primary cultures? Although, it is convenient to use fibroblasts, this cell type does not represent a tissue that is extensively affected in these pathologies. This point is not discussed. The Table 3 should be converted to a bar plot. Regarding GO analysis, the entities selected are redundant and confusing. An analysis with an FDR < 0.1 does not seem relevant, a threshold <0.05 would be appropriate. The authors should provide the full statistical output with fold changes, p value and adjust p value for each gene as supplementary material. Table 4 is not easily digestible. I strongly recommend to convert it two separate heatmaps of upregulated and dowregulated genes Figure 1 is not readable and confusing. It should be replaced by a new figure with better annotations such as Venn diagram or Chord diagram Table 6 please sort the gene column by upregulated and downregulated status to enhance the table clarity. The authors suggest that the impairment of different cellular processes explain the failures of enzyme replacement therapy without taking into consideration the bioavailability of the recombinant enzymes. The fact that the enzymes do not cross the BBB and that the bones are poorly irrigated organs also explain why ERT does not relieve all the symptoms of MPS.
Minor points:
- Please select the most relevant cellular processes for table 4 to fit it into a single page.
- In the Material and Methods section, gene names should be in italic.
- MPS types are sometimes designated using Arabic numerals (supplementary figure S1 heatmaps and table S1) as opposed to Roman numerals (main manuscript, supplementary table S2 and tables in figure S1).
- There is a problem in the page numbering (pages 8 to 17 are numbered from 1 to 10).
Author Response
Reviewer #2
We thank the reviewer for constructive comments. Changes in the manuscript, introduced in response to the review, are marked in blue.
REVIEWER’S COMMENT: The authors stated that they used four biological repeats for the experiments, but they do not detail how they obtained these biological replicates.
RESPONSE: The explanation how four biological replicates were performed is now provided in the revised manuscript (line 153).
REVIEWER’S COMMENT: Besides, studying only one sample per pathology represents a serious limitation that has not been discussed.
RESPONSE: This limitation is now discussed in the revised manuscript (lines 368-382).
REVIEWER’S COMMENT: In addition, there are inconstancies in gene expression in the biological repeats and this point is not discussed.
RESPONSE: Differences in results between particular biological repeats are common in this type of experiments. Nevertheless, we thank the reviewer for putting attention on this problem, and we fully agree that it should be discussed. We have introduced such a discussion into the revised text (lines 382-386).
REVIEWER’S COMMENT: Are the fibroblasts derived from cell lines or from primary cultures?
RESPONSE: The fibroblasts were cell lines, as indicated in the revised manuscript (line 99).
REVIEWER’S COMMENT: Although, it is convenient to use fibroblasts, this cell type does not represent a tissue that is extensively affected in these pathologies. This point is not discussed.
RESPONSE: The advantages and disadvantages of the use of fibroblasts in our studies are discussed in the revised manuscript (lines 387-396).
REVIEWER’S COMMENT: The Table 3 should be converted to a bar plot.
RESPONSE: As requested by the reviewer, this table has been replaced with a bar plot (Figure 1 in the revised manuscript).
REVIEWER’S COMMENT: Regarding GO analysis, the entities selected are redundant and confusing. An analysis with an FDR < 0.1 does not seem relevant, a threshold <0.05 would be appropriate.
RESPONSE: Although we agree that in vast majority of biological and medical experiments the threshold of significance is assumed to be <0.05, in bioinformatic analyses, especially in large-scale experimental data (like transcriptomic studies) it is commonly accepted to use FDR<0.1 as a standard. Examples of such analyses, published in recognized journals, can be find in articles listed below. Nevertheless, we have also performed analysis with significantly more strict threshold, <10e-6, and in fact most conclusions presented in this report are based on such analyses. Please, find examples of articles in which FDR<0.1 was used as a threshold in transcriptomic analyses:
Galbraith DA, Yang X, Niño EL, Yi S, Grozinger C. Parallel epigenomic and transcriptomic responses to viral infection in honey bees (Apis mellifera). PLoS Pathog. 2015; 11(3):e1004713. doi: 10.1371/journal.ppat.1004713.
Solano-Aguilar G, Molokin A, Botelho C, Fiorino AM, Vinyard B, Li R, Chen C, Urban J Jr, Dawson H, Andreyeva I, Haverkamp M, Hibberd PL. Transcriptomic profile of whole blood cells from elderly subjects fed probiotic bacteria Lactobacillus rhamnosus GG ATCC 53103 (LGG) in a phase I open label study. PLoS One. 2016;11(2):e0147426. doi: 10.1371/journal.pone.0147426.
Matkovich SJ, Grubb DR, McMullen JR, Woodcock EA. Chronic contractile dysfunction without hypertrophy does not provoke a compensatory transcriptional response in mouse hearts. PLoS One. 2016;11(6):e0158317. doi: 10.1371/journal.pone.0158317.
REVIEWER’S COMMENT: The authors should provide the full statistical output with fold changes, p value and adjust p value for each gene as supplementary material.
RESPONSE: As requested by the reviewer, full statistical analyses for each relevant gene is provided in supplementary material now.
REVIEWER’S COMMENT: Table 4 is not easily digestible. I strongly recommend to convert it two separate heatmaps of upregulated and dowregulated genes
RESPONSE: As suggested by the reviewer, results presented previously in Table 4 are show in the form o plots (Figure 2) in the revised manuscript. We have chosen the form of bar plots as detailed heat maps are presented in supplementary material, thus, we aimed to avid repetition of the presented material.
REVIEWER’S COMMENT: Figure 1 is not readable and confusing. It should be replaced by a new figure with better annotations such as Venn diagram or Chord diagram
RESPONSE: We agree that Figure 1A might be misleading. Therefore, we have replaced it with a new version which is easy to follow. Moreover, we have presented the results in the form of Chord graph, as a new Figure 4, and described the analysis in the text (lines 206-274, 431-442).
REVIEWER’S COMMENT: Table 6 please sort the gene column by upregulated and downregulated status to enhance the table clarity.
RESPONSE: As requested by the reviewer, the genes in the table (Table 4 in the revised manuscript) have been sorted by upregulation and downregulation status.
REVIEWER’S COMMENT: The authors suggest that the impairment of different cellular processes explain the failures of enzyme replacement therapy without taking into consideration the bioavailability of the recombinant enzymes. The fact that the enzymes do not cross the BBB and that the bones are poorly irrigated organs also explain why ERT does not relieve all the symptoms of MPS.
RESPONSE: According to reviewer’s recommendation, we have discussed the features of ERT which significantly influence the efficacy of this therapy in CNS and bones (lines 63-65, 556-558).
Minor points:
REVIEWER’S COMMENT: Please select the most relevant cellular processes for table 4 to fit it into a single page.
RESPONSE: Former Table 4 is now presented in the form of a graph which fits into a single page (Figure 2).
REVIEWER’S COMMENT: In the Material and Methods section, gene names should be in italic.
RESPONSE: Names of genes are in italic font now.
REVIEWER’S COMMENT: MPS types are sometimes designated using Arabic numerals (supplementary figure S1 heatmaps and table S1) as opposed to Roman numerals (main manuscript, supplementary table S2 and tables in figure S1).
RESPONSE: We thank reviewer for this note. We are aware that in supplementary Figure S1 heatmaps and Table S1 the names of MPS types are presented in Arabic numerals which is not a standard in MPS nomenclature. However, we would like to keep this designation in this supplementary material, as due to complexity of the presented data (including numbers of biological repeats), if we used Roman numerals, the names of MPS types became unclear and almost unreadable which might be confusing for readers. In the revised supplementary material, we have provided explanation for this unconventional nomenclature.
REVIEWER’S COMMENT: There is a problem in the page numbering (pages 8 to 17 are numbered from 1 to 10).
RESPONSE: Page numbering has been corrected.
Round 2
Reviewer 2 Report
This version takes into account the proposed changes and the results are better presented and valued.
Minor changes may improve this manuscript.
Table 2: to make the genotype more readable, the column ''Mutation'' could be replaced by the columns below
| Gene identification | Variant 1 | Variant 2 | ||
| Nucleotide change | Protein change | Nucleotide change | Protein change | |
| NM_XXXXXX | p.Trp402* | p.Trp402* | ||
Author Response
Table 2 has been modified accorind to reviewer's recommendation.